# Low T Cell Responsiveness in the Early Phase of COVID-19 Associates with Progression to Severe Pneumonia in Kidney Transplant Recipients

**DOI:** 10.3390/v14030542

**Published:** 2022-03-05

**Authors:** Marion Cremoni, Sébastien Cuozzo, Emanuela Martinuzzi, Susana Barbosa, Nadia Ben Hassen, Filippo Massa, Elisa Demonchy, Matthieu Durand, Olivier Thaunat, Vincent Esnault, Moglie Le Quintrec, Sophie Caillard, Nicolas Glaichenhaus, Antoine Sicard

**Affiliations:** 1Department of Nephrology Dialysis and Transplantation, Pasteur 2 Hospital, Nice University Hospital, 06001 Nice, France; cremoni-gauci.m@chu-nice.fr (M.C.); cuozzo.s@chu-nice.fr (S.C.); benhassen-dakhlaoui.n@chu-nice.fr (N.B.H.); esnault.v@chu-nice.fr (V.E.); 2Clinical Research Unit Côte d’Azur (UR2CA), University Côte d’Azur, 06200 Nice, France; 3Institute of Molecular and Cellular Pharmacology, University Côte d’Azur, CNRS, 06560 Valbonne, France; martinuzzi@ipmc.cnrs.fr (E.M.); sudocarmo@gmail.com (S.B.); glaichenhaus@ipmc.cnrs.fr (N.G.); 4Laboratory of Molecular Physio Medicine, University Côte d’Azur, CNRS, 06107 Nice, France; filippo.massa@univ-cotedazur.fr; 5Infectious Diseases Department, Archet 1 Hospital, Nice University Hospital, 06200 Nice, France; demonchy.e@chu-nice.fr; 6Urology Department, Pasteur 2 Hospital, Nice University Hospital, 06001 Nice, France; durand.m@chu-nice.fr; 7Department of Transplantation, Nephrology and Clinical Immunology, Edouard Herriot Hospital, Hospices Civils de Lyon, 69003 Lyon, France; olivier.thaunat@inserm.fr; 8Department of Nephrology, Dialysis and Renal Transplantation, University Hospital of Lapeyronie, 34090 Montpellier, France; m-lequintrec-donnette@chu-montpellier.fr; 9Department of Nephrology and Transplantation, Strasbourg University Hospital, 67091 Strasbourg, France; sophie.ohlmann@chru-strasbourg.fr

**Keywords:** kidney transplantation, COVID-19, SARS-CoV-2, T lymphocytes, immune responsiveness

## Abstract

Kidney transplant (KT) recipients are at increased risk of developing severe forms of COVID-19. Little is known about the immunological mechanisms underlying disease severity in these patients receiving T-cell targeting immunosuppressive drugs. We investigated the relationship between T cell responsiveness at the beginning of the infection and the risk of subsequent progression to respiratory failure. We performed a multicentric prospective study in KT recipients with a positive RT-PCR COVID-19 test and only mild symptoms at inclusion. Blood samples were collected at baseline in a cell culture system containing T cell stimuli. We assessed T cell responsiveness by computing the ratio between the levels of Th1, Th2, Th17 and Treg cytokines produced after polyclonal stimulation and the number of blood lymphocytes. We then used an unsupervised classification approach to stratify patients into low and high T cell responders and a penalized logistic regression to evaluate the association between T cell responsiveness and progression to severe pneumonia. Forty-five patients were included. All patients who progressed to severe pneumonia (24.4%, n = 11) were low T cell responders at baseline (*p* = 0.01). In multivariate analysis, low T cell responsiveness at baseline was the main risk factor for subsequent progression to severe pneumonia. This study provides novel insights into the mechanisms underlying COVID-19 severity in organ transplant recipients and data of interest to clinicians managing immunosuppressive drugs in these patients.

## 1. Introduction

Kidney transplant (KT) recipients with COVID-19 are more prone to developing severe respiratory symptoms and death compared to the general population [1,2,3,4,5,6,7,8,9]. However, COVID-19 leads to very different outcomes in kidney transplant patients. While some patients recover in a few days, a significant proportion of patients develop life-threatening respiratory failure. Understanding the mechanisms leading to respiratory deterioration in these patients is essential to improve their management and decrease COVID-19 mortality in this high-risk population. Although KT recipients are severely immunocompromised, little is known about the immunopathogenesis of COVID-19 in this population. In the general population, it was shown that patients with severe forms of COVID-19 requiring hospitalization and intensive care had impaired T cell responses compared to patients with benign infection [10,11,12]. In patients with mild symptoms at the beginning of the infection, no relationship between T cell responses and the risk to progress to a severe form of COVID-19 was reported in the general population. KT recipients differ from the general population in that they have altered T cells responses due to their immunosuppressive therapy that includes drugs targeting T cells such as corticosteroids, calcineurins inhibitors, antimetabolites, mTOR inhibitors and/or Belatacept. In order to improve the understanding of the immunopathogenesis of COVID-19 in kidney transplant recipients, we studied the relationship between T cell responsiveness measured in an early phase of COVID-19 in KT recipients with mild symptoms and the risk to subsequently progress to severe pneumonia.

## 2. Materials and Methods

### 2.1. Study Design, Participants, Data Collection

This study is a national multicentric prospective cohort study. Fifty-two KT recipients with RT-PCR-documented COVID-19 were included between 8 September 2020 and 22 March 2021 in the University Hospitals of Nice, Strasbourg, Montpellier and Lyon, France. Non-inclusion criteria were: onset of symptoms for more than 10 days; signs of severe disease (oxygen therapy > 3 L/min, blood pressure < 85/55 mmHg, hemodynamic instability, encephalopathy); treatment with high dose corticosteroids within the last 14 days preceding inclusion; and documented active bacterial or fungal infection. Seven patients were excluded from the study because they had received high-dose corticosteroids before blood collection (Appendix A). Progression to severe pneumonia was defined as the development of severe hypoxemia (>4 L/min) requiring dexamethasone administration, admission to an intensive care unit and/or leading to death. Demographic, clinical, laboratory and outcome data were collected by the investigating physician using the electronic observation booklet of the study. It should be noted that at the time of the study, no patient had been vaccinated against SARS-CoV-2.

### 2.2. Collection of Blood Samples and Immunoassays

Blood sampling was performed at inclusion. For assessing immune cell responsiveness, one milliliter of blood was drawn into an integrated collection and cell culture system (TruCulture^®^, Myriad RBM, Austin, TX, USA), with or without anti-CD3 and anti-CD28 agonist monoclonal antibodies (mAbs) for T cell activation. TruCulture^®^ tubes were incubated for 25 h (±15 min) at 37 °C using a bench-top heating block (VLMH GmbH, Wien, Austria). After incubation, a valve separator was inserted into the tubes allowing for the collection of cellular supernatants that were stored at −80 °C. The concentrations of interleukin (IL)-2, IL-5, IL-10, IL-17A, interferon (IFN)-γ and granulocyte-macrophage colony-stimulating factor (GM-CSF) in supernatants were measured using the V-PLEX^®^ Cytokine Panel 1 kit (MesoScaleDiscovery, MD, USA). Data were acquired on the V-PLEX^®^ Sector Imager 2400 plate reader and analyzed using the Discovery Workbench 3.0 software (MesoScaleDiscovery, MD, USA).

### 2.3. Statistical Analysis

#### 2.3.1. Descriptive Statistics

Data are presented as mean and standard deviation (SD), median and range (minimum–maximum), or as counts and percentages. The Shapiro–Wilk normality test was used to verify the distribution of data. Comparisons were performed using the unpaired two-sided Student’s t-test, Wilcoxon–Mann–Whitney U test, or Fisher’s exact test as appropriate. Statistical analyses were performed using GraphPad Prism 8.0 (GraphPad Software, Inc., San Diego, CA, USA). Differences were considered significant when *p*-value < 0.05.

#### 2.3.2. Unsupervised Classification

The unsupervised classification was performed using the k-spectral clustering algorithm, aiming at identifying two homogeneous clusters within our study sample based on cytokine levels normalized to lymphocyte numbers. Silhouette values were computed as a measure of clustering appropriateness. We used the t-Distributed Stochastic Neighbor Embedding (t-SNE) method for the graphical representation of the clusters.

#### 2.3.3. Penalized Logistic Regression

The caret [13] and glmnet [14] R packages were used to implement penalized logistic regression models with the aim of performing variable selection, leading to a sparser final model. We implemented the penalized logistic regression over 200 non-parametric resampling bootstraps. We calculated a Variable Inclusion Probability (VIP) and the proportion of bootstrap runs out of the 200 in which a given variable was kept in the model. In the absence of asymptotically valid p-values, which are not available in high-dimensional regression, the VIP can be interpreted as the posterior probability of including a variable in the model and is used as a measure of the stability of the association with the outcome [15]. The use of a conservative threshold of 50% is recommended if the goal is not to miss any possibly relevant predictors [15]. However, we used a VIP threshold of 95% in this study to only identify stable associations and avoid false positives, which was consistent with reference work in Bayesian Statistics [16].

## 3. Results

### 3.1. Characteristics of the Study Population and Outcomes

Forty-five kidney transplant recipients with a SARS-CoV-2 infection and only mild symptoms were enrolled in the study. The patients’ characteristics are shown in Table 1. Patients were infected by SARS-CoV-2 a median of 35 (1–400) months following kidney transplantation. Six (13%) patients were infected within 3 months of transplantation. The most frequent comorbidities were hypertension (91%), obesity (22%) and diabetes (20%). Baseline creatininemia and estimated glomerular filtration rate (eGFR) were 127.5 (79.6–460.0) µmol/L and 49.0 (13.0–120.0) mL/min per 1.73m², respectively. Maintenance immunosuppressive therapy at inclusion included calcineurin inhibitors (96%), antimetabolites (82%) and corticosteroids (76%, baseline dose between 5 and 10 mg/day). No patient had experienced clinical or subclinical rejection episodes before inclusion. COVID-19 symptoms began 5.8 (SD, 2.1) days before inclusion. Common symptoms were reported by the majority of patients. Baseline immunosuppression was modified after inclusion in 21 (47%) patients: antimetabolites were stopped in 20 patients, and calcineurin inhibitors were stopped in one patient. None of the patients received antiviral treatment. Among the 45 patients of the study, 11 (24%) subsequently progressed to severe pneumonia requiring hospitalization and high dose dexamethasone, of whom two were transferred to an intensive care unit, and one died. There was no graft loss.

### 3.2. Patients Stratification Based on Early T Cells Responsiveness

In order to evaluate T cell responsiveness at baseline, we collected blood at inclusion into an integrated collection and cell culture system with or without anti-CD3/CD28 agonist monoclonal antibodies (mAbs). After 25 h of culture, we assessed cellular supernatants for cytokines secreted by T helper (Th)1 cells (IL-2, IFN-γ and GM-CSF), Th2 cells (IL-5), Th17 cells (IL-17A) and regulatory T cells (IL-10). We then computed the ratio between cytokine concentration and the number of blood lymphocytes. As expected, anti-CD3/CD28 mAbs induced the secretion of IL-2, IL-5, IL-10, IL-17A, IFN-γ and GM-CSF (Appendix A). An unsupervised classification approach based on the levels of IL-2, IL-5, IL-10, IL-17A, IFN-γ and GM-CSF produced in response to anti-CD3/CD28 mAbs resulted in two clusters, cluster 1 and cluster 2 that consisted of 14 and 31 patients, respectively. In agreement with the relatively high mean silhouette value of the clustering solution, i.e., 0.82, patients from the two clusters projected in distinct regions of t-distributed Stochastic Neighbor Embedding (t-SNE) graphical representation reflecting high intra-cluster consistency (Figure 1). Compared to cells from cluster 1 patients, those from cluster 2 secreted less IL-2, IL-5, IL-10, IL-17A, IFN-γ and GM-CSF in response to T cell stimuli (Figure 2). In summary, we stratified patients into two groups of high and low T cell responders that consisted of 14 and 31 patients, respectively.

### 3.3. Characteristics and Outcomes of Low and High T Cell Responders

Patients’ characteristics at baseline were similar in high and low T cell responders (Table 2). As for the outcome, none (0 out of 14) of the high T cell responders progressed to severe pneumonia while 35% (11 out of 31) of low T cell responders did.

### 3.4. Variables Associated with the Risk to Progression to Severe Pneumonia

We next performed a penalized logistic regression to investigate the association between clinical outcome and T cell responsiveness after adjustment for age, sex, body mass index (BMI), time since transplantation, diabetes, basal creatininemia, treatment with calcineurin inhibitors, mTOR inhibitors, steroids and antimetabolites. Using a VIP threshold of 95% indicative of a strong association between the outcome and the independent variables, we found that low T cell responsiveness (OR 3.00 (1.53–39.10), VIP 0.98) and high basal creatininemia (OR 2.06 (1.13–9.21), VIP 0.97), were independently associated with an increased risk of progression to severe pneumonia. The absence of antimetabolite treatment was associated with a significantly lower risk of progression to severe pneumonia (OR 0.42 (0.06–0.91), VIP 0.95). The results are presented in Table 3.

## 4. Discussion

Kidney transplant recipients are severely immunocompromised and have a high risk of developing severe forms of COVID-19. However, little is known about the immunological mechanisms underlying disease severity in this population. Here, we prospectively assessed the relationship between early immune responsiveness in KT recipients with COVID-19 and the risk of progressing to severe SARS-CoV-2 pneumonia. We show that low T cell responsiveness in the early phase of the infection is a major risk factor for COVID-19 progression in this population.

Previous important studies analyzed the risk factors for poor COVID-19 outcomes in kidney transplant recipients [2,3,4,5,17,18,19,20,21]. These studies included heterogeneous populations. In contrast, our prospective study was focused on patients who exhibited only mild symptoms in the first days of SARS-CoV-2 infection. While most of the studies did not include any functional immunoassay, we used a reproducible blood collection and culture system to measure immune cell actability [22,23,24,25] and investigate the immunopathogenesis of COVID-19 in a homogeneous population of KT recipients.

Immunosuppressive therapy is a major determinant of T cell responsiveness in KT recipients and may therefore impact COVID-19 outcomes. The maintenance immunosuppressive therapy can differ significantly between KT recipients. The choice of immunosuppressive drugs depends on the practices of the transplantation centers and on patients’ characteristics such as age, number of previous transplantations, time since transplantation, the individual risk of rejection or infection. Immunosuppressive drugs also display high interindividual variability in pharmacokinetics and pharmacodynamics [26,27,28]. The maintenance immunosuppressive therapy may therefore have a different impact on T cells blockage at the individual level and may differently influence COVID-19 outcomes. In our cohort, we identified antimetabolites as a risk factor for progression to severe COVID-19 pneumonia. This observation is coherent with recent studies, which showed that poor response to COVID-19 vaccination was persistently associated with the use of antimetabolite immunosuppression in organ transplant recipients [29].

The SARS-CoV-2 virus is able to modulate host immune responses in some patients, and severe forms of COVID-19 have been associated with impaired T cell immunity [30]. However, no association between T cell responses and outcomes was reported in recently infected patients with mild symptoms in the general population, which contrasts with our results. This may be explained by the fact that a majority of individuals in the general population are immunocompetent and have globally functional and comparable T cell responses at baseline. In contrast, KT recipients have impaired and qualitatively and quantitatively heterogeneous T cells responses. Differences in baseline immune responsiveness in KT patients may induce different susceptibilities to SARS-CoV-2 immune manipulations. Our results suggest that lowering immunosuppressive therapy at the time of COVID-19 diagnosis in these patients may improve T cell responsiveness, decrease susceptibility to SARS-CoV-2 and facilitate the implementation of an adaptive immune response. However, immunosuppression minimization has to be balanced against the risk of rejection and can only be envisaged by the clinician in charge of the patient on a case-by-case basis.

The main limitations of this study are the relatively small sample size, the absence of assessment of SARS-CoV-2-specific T cell responses, and the lack of a control group of non-immunosuppressed subjects. However, robust and advanced statistical methods were used to account for the population size, and antigen-specific T cell responses were too difficult to detect in the first days of the infection in this immunocompromised population.

In conclusion, this study demonstrates that KT recipients who have low T cell responsiveness in the early phase of COVID-19 have a higher risk of developing a life-threatening form of the disease and provides new insights into the immunopathogenesis of COVID-19 in kidney transplant recipients and important data for clinicians managing immunosuppressive drugs in these patients.

## Figures and Tables

**Figure 1 viruses-14-00542-f001:**
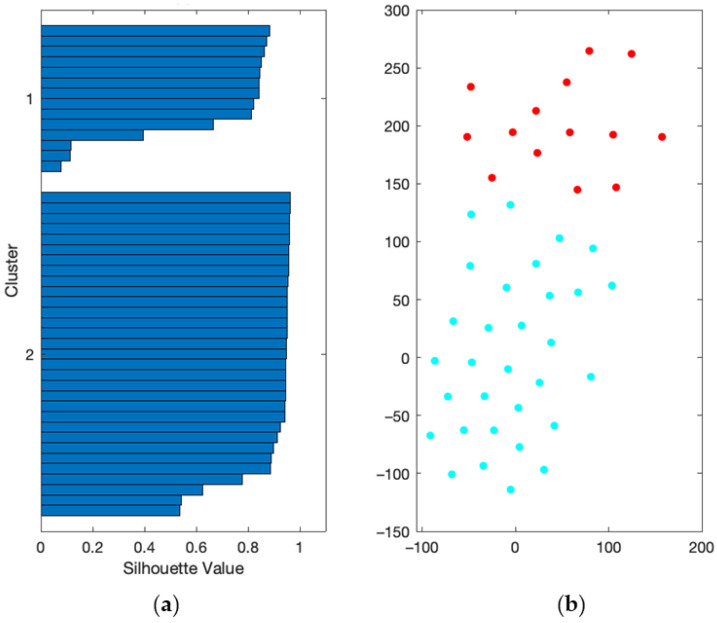
Unsupervised classification performance. (**a**) Silhouette values, a measure of the individual’s parenthood within the underlying cluster, were plotted on the y-axis for high (cluster 1) and low (cluster 2) T cell responders, respectively. (**b**) t-SNE graphical model of k-spectral clustering analysis with two clusters. Each dot represents an individual projected in a two-dimensional space. High T cell responders are represented by a red dot, and low T cell responders by a blue dot.

**Figure 2 viruses-14-00542-f002:**
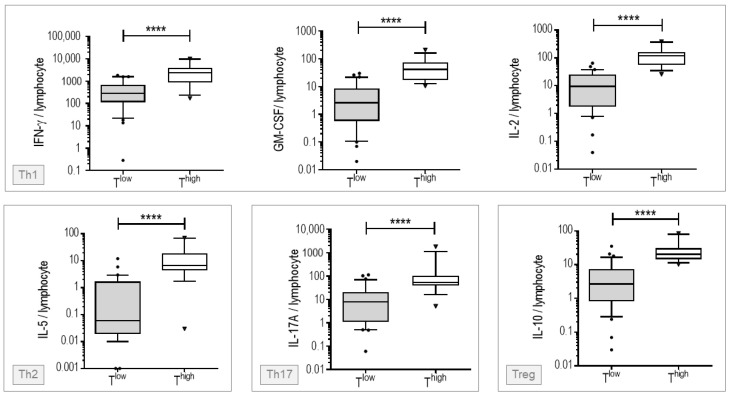
Cytokine levels secreted by blood cells of low and high T cell responders. Blood cells were stimulated at 37 °C for 25 h with anti-CD3/CD28 agonist mAbs in an integrated collection and cell culture system. IL-2, IL-5, IL-10, IL-17A, IFN-γ and GM-GSF levels were measured in cellular supernatants. The ratio between the levels of cytokines and the number of blood lymphocytes was computed. Statistical significance of differences between groups was assessed using the Mann–Whitney non-parametric test. T^high^, high T cell responders; T^low^, low T cell responders. **** *p* < 0.0001.

**Table 1 viruses-14-00542-t001:** Patients’ characteristics at baseline.

**Baseline characteristics**	
Age, years	51.7 (16.9)
Males	33 (73%)
Comorbidities	
Diabetes	9 (20%)
Hypertension	41 (91%)
Cardiovascular events	4 (9%)
BMI > 30 kg/m²	10 (22%)
COPD	1 (2%)
Time since transplantation, months	35 (1–400)
Prior renal transplantation	4 (9%)
Pretransplant Donor Specific Antibodies	0 (0%)
Baseline creatininemia, µmol/L	127.5 (79.6–460.0)
Baseline eGFR, mL/min/1.73 m²	49.0 (13.0–120.0)
**Immunosuppressive therapy at inclusion**	
Corticosteroids	34 (76%)
Calcineurin inhibitors	43 (96%)
Antimetabolites	37 (82%)
mTOR inhibitors	3 (7%)
**Presentation at inclusion**	
Time since symptoms onset, days	5.8 (2.1)
Fever	11 (24%)
Cough	16 (36%)
Dyspnea with normal oxygen saturation	8 (18%)
Anosmia/ageusia	11 (24%)
Diarrhea	8 (18%)
Headache	16 (36%)
Serum creatinine, µmol/L	142.1 (79.5–398.0)
Lymphocytes count, ×10^9^/L	0.9 (0.2–3.7)
Residual tacrolemia, µg/L *	7.19 (2.30–18.40)
**Clinical course**	
Progression to severe pneumonia	11 (24%)
Subsequent ICU admission	2 (4%)
Death	1 (2%)

The number (n) and percentage (%) of patients are indicated for categorical variables: sex, comorbidities (diabetes, hypertension, cardiovascular events, obesity), prior renal transplantation, treatment with corticosteroids, calcineurin inhibitors, antimetabolites and mTOR inhibitors, clinical symptoms (fever, cough, dyspnea with normal oxygen saturation, anosmia and/or ageusia, diarrhea, headache) and clinical outcome (progression to severe pneumonia, intensive care unit admission, death). Mean and standard deviation (SD), or median and range (minimum–maximum), are shown for continuous variables as appropriate: age (years), time since transplantation (years), baseline creatininemia (µmol/L), baseline eGFR (mL/min/1.73 m²), time since symptoms onset (days), serum creatinine levels (µmol/L), lymphocytes count (× 10^9^/L), residual tacrolemia (µg/L). COPD—chronic obstructive pulmonary disease; ICU—intensive care unit.* 37/45 patients were treated with tacrolimus.

**Table 2 viruses-14-00542-t002:** Sample characteristics of low and high T cell responders.

	Low T cell responders (n = 31)	High T cell responders (n = 14)	*p*-value
**Baseline characteristics**			
Age, years	53.0 (17.3)	48.9 (16.1)	0.45
Males	22 (71%)	11 (79%)	0.73
Comorbidities			
Diabetes	7 (23%)	2 (14%)	0.70
Hypertension	27 (87%)	14 (100%)	0.29
Cardiovascular events	2 (6%)	2 (14%)	0.58
BMI> 30 kg/m²	7 (23%)	3 (21%)	>0.99
Time since transplantation, months	21 (1–400)	57 (10–186)	0.13
Prior renal transplantation	2 (6%)	2 (14%)	0.58
Baseline creatininemia, µmol/L	120.0 (79.6–460.0)	145.5 (94.0–195.0)	0.42
Baseline eGFR, mL/min/1.73 m²	53.0 (13.0–64.0)	43.5 (28.0–85.0)	0.57
**Immunosuppressive therapy at inclusion**			
Corticosteroids	23 (74%)	11 (79%)	>0.99
Calcineurin inhibitors	30 (97%)	13 (93%)	0.53
Antimetabolites	28 (90%)	9 (64%)	0.08
mTOR inhibitors	2 (6%)	1 (7%)	>0.99
**Presentation at inclusion**			
Time since symptoms onset, days	5.8 (2.1)	5.6 (2.2)	0.78
Fever	9 (29%)	2 (14%)	0.46
Cough	11 (35%)	5 (36%)	>0.99
Dyspnea with normal oxygen saturation	6 (19%)	2 (14%)	>0.99
Anosmia/ageusia	7 (23%)	4 (29%)	0.72
Diarrhea	5 (16%)	3 (21%)	0.69
Headache	9 (29%)	7 (50%)	0.20
Serum creatinine, µmol/L	142.1 (79.5–398.0)	141.4 (94.0–231.8)	0.67
Lymphocytes count, ×10^9^/L	0.8 (0.2–3.7)	1.1 (0.3–2.9)	0.31
Residual tacrolemia, µg/L *	9.05 (4.46)	7.26 (1.66)	0.50
**Immunosuppression management**			
Antimetabolites withdrawal	16/28 (57%)	4/9 (44%)	0.70
**Clinical course**			
Progression to severe pneumonia	11 (35%)	0 (0%)	**0.01**

Categorial (number and percentage on subgroup) and continuous variables (mean and SD, or median and range) listed in Table 1 are shown for low and high T cell responders, respectively. Low and high T cell responders characteristics were compared using univariate Chi-square or Fisher’s exact test and Mann–Whitney U test for categorical and continuous variables, respectively. Differences between groups were considered to be statistically significant when corrected *p*-values were < 0.05. Significant associations are highlighted. * 37/45 patients were treated with tacrolimus.

**Table 3 viruses-14-00542-t003:** Association between clinical outcome and patient T cell responsiveness.

	Median ORs	%95 CI	VIP
High T cell responsiveness (cluster 1)	0.43	[0.05–1.00]	0.85
Low T cell responsiveness (cluster 2)	3.00	[1.53–39.10]	**0.98**
Age (years)	0.86	[0.12–1.87]	0.58
Sex (Female)	0.63	[0.13–2.31]	0.65
Sex (Male)	1.49	[0.47–6.72]	0.62
BMI (kg/m²)	1.28	[0.48–4.32]	0.68
Time since transplantation (years)	0.69	[0.14–1.40]	0.68
Basal creatinemia (µmoL/L)	2.06	[1.13–9.21]	**0.97**
Diabetes (No)	0.43	[0.02–1.49]	0.84
Diabetes (Yes)	1.73	[0.66–33.09]	0.75
Calcineurin inhibitor (No)	0.52	[0.08–0.95]	0.57
Calcineurin inhibitor (Yes)	1.83	[1.00–10.18]	0.53
mTOR inhibitor (No)	1.13	[0.32–3.58]	0.37
mTOR inhibitor (Yes)	0.97	[0.37–2.90]	0.37
Steroids (No)	1.47	[0.48–11.63]	0.70
Steroids (Yes)	0.84	[0.10–2.08]	0.67
Antimetabolites (No)	0.42	[0.06–0.91]	**0.95**
Antimetabolites (Yes)	1.92	[1.00–10.16]	0.89

Penalized logistic regressions were fitted using the clinical outcome as the dependent variable and T cell responsiveness (low vs. high) as an independent variable with other confounders. Median Odds Ratios (ORs), 95% confidence intervals (CI) and VIP are shown. Strong associations are highlighted.

## Data Availability

Data supporting the conclusions of this study are available from the first and last authors.

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
