# Peer review of "Low T Cell Responsiveness in the Early Phase of COVID-19 Associates with Progression to Severe Pneumonia in Kidney Transplant Recipients"

_viruses, 2022, doi:10.3390/v14030542_

Round 1

Reviewer 1 Report

Authors have evaluated a small number of renal transplant patients on the basis of their T cell responsiveness to COVID 19 infection and subsequently the clinical outcomes. Overall, presentation of the paper is satisfactory with a novel insight into the variation of clinical response of different patients to COVID 19 infection. However, few important questions remained unanswered-

  1. how was the immunosuppression managed once the diagnosis of COVID 19 was made?
  2. Were patient’s other comorbidities related to pulmonary function like baseline COPD, Asthma etc recorded? As these may have impact on the outcome.
  3. What was the baseline dose of steroids in these patients?
  4. what was the treatment given for COVID 19 infection in mild cases?
  5.  

Author Response

We thank the reviewer for his evaluation of our work and his comments which have allowed us to improve our manuscript.

« How was the immunosuppression managed once the diagnosis of COVID 19 was made? »

Baseline immunosuppression was modified after inclusion in 21 (47%) patients: antimetabolites were stopped in 20 patients and calcineurin inhibitors were stopped in one patient. This information has been added to the revised version of the manuscript (results section, line 136-138).

« Were patient’s other comorbidities related to pulmonary function like baseline COPD, Asthma etc recorded? As these may have impact on the outcome. »

Comorbidities related to pulmonary function were rare: 1 patient had COPD, none had asthma, none had cancer. This information has been added in the revised version of the manuscript (table 1)

Of note, other comorbidities were also rare in this cohort: no patient had cirrhosis, 2 had inflammatory or autoimmune disease (1 ankylosing spondylitis and 1 unspecified), and 1 patient had chronic inflammatory bowel disease.

« What was the baseline dose of steroids in these patients? »

The baseline dose of steroids was between 5mg and 10 mg/day. This information has been added in the revised version of the manuscript (results section, line 133).

« What was the treatment given for COVID 19 infection in mild cases? »

None of the patients received anti-viral treatment. This information has been added in the revised version of the manuscript (results section, line 138).

Reviewer 2 Report

Paper seems well built and focuses on an important aspect in the approach of a transplanted patient affected by Covid 19 infection, however it seems to hesitate in giving practical indications on the therapeutic conduct to be adopted at the time of the diagnosis of Covid 19.

In addition, some details need to be specified in order to complete the description of the patients:

Standard values of the T cell response in a group of non-immunosuppressed subjects

The distance from the transplant should be specified, in table 1 the minimum seems 1.1 years, in table 2, in the LOW group a 0.3 appears which seems inconsistent with the data provided in table 1. In any case should be included only patients with stabilized therapy for at least 3 months including additional treatments for post-operative rejection episodes that require immunosuppressive enhancement.

Similar reasoning for any long-acting immunosuppressant

The pre-transplant immunization status (PRA) and any more aggressive therapeutic approach (if done) could be interesting

It should be specified whether patients with signs of rejection at protocol biopsies were included or excluded

It seems to understand that the T cell response tests were carried out at the time of the diagnosis of Covid 19 to patients still on non-reduced therapy

It would be interesting to know if immunosuppression was reduced in all patients and with which drug (MMF?). In addition it would be interesting if this was associated with a different evolution from infection to disease

It would be interesting to isolate the population that did not take basicly antimetabolites (3 in the LOW group and 5 in the HIGH group) and look not only at the T cell response but if this best response correlates with the best outcome of the Covid infection

It is not described if there have been deaths or loss of the graft (for example due to thrombophilic problems)

Author Response

“Paper seems well built and focuses on an important aspect in the approach of a transplanted patient affected by Covid 19 infection”

We thank the reviewer for his positive evaluation of our work and constructive comments that have allowed us to improve our manuscript.

“However it seems to hesitate in giving practical indications on the therapeutic conduct to be adopted at the time of the diagnosis of Covid 19.”

Because of the relatively low study sample size and because the study was not interventional, we decided to be careful with our conclusions. To insist on the practical implications of our results as suggested by the reviewer, the discussion section has been revised : “Our results suggest that lowering immunosuppressive therapy at time of COVID-19 diagnosis in these patients may improve T cell responsiveness, decrease susceptibility to SARS-CoV-2 and facilitate the implementation of an adaptive immune response.” (discussion section, line 254-257).

« Standard values of the T cell response in a group of non-immunosuppressed subjects »

We agree with the reviewer that this would have been very interesting to compare the values of the T cell response in transplant patients and those in non-immunosuppressed subjects. Unfortunately, non-immunosuppressed subjects were not included initially in the study design, and our ethical protocol does not allow us to include a control group of healthy subjects. The time required to obtain ethical authorization is not compatible with our deadlines. A commentary on this important point has been added in the discussion section of the study (limitations paragraph of the discussion section, line 261-262).

« The distance from the transplant should be specified, in table 1 the minimum seems 1.1 years, in table 2, in the LOW group a 0.3 appears which seems inconsistent with the data provided in table 1. In any case should be included only patients with stabilized therapy for at least 3 months including additional treatments for post-operative rejection episodes that require immunosuppressive enhancement. Similar reasoning for any long-acting immunosuppressant. The pre-transplant immunization status (PRA) and any more aggressive therapeutic approach (if done) could be interesting »

We apologize for the lack of clarity. In the first version of the manuscript, data concerning the distance from the transplant were presented as median and interquartiles [25th percentile - 75th percentile] and not as median and ranges [min-max]. In the revised version of the manuscript, all data are now presented as median and ranges [min - max] and all units are in months to improve clarity. Time since transplantation has been included in the multivariable analysis to avoid bias related to the distance from the transplant. The pre-transplant immunization status (pretransplant DSA) has been added in table 1 in the revised version of the manuscript. No patient of the study received aggressive immunosuppressive treatment (like desensitization) before inclusion.

« It should be specified whether patients with signs of rejection at protocol biopsies were included or excluded »

No patient with signs of rejection on protocol biopsies was included in the study. This information has been added in the revised version of the manuscript as suggested by the reviewer (result section, line 134).

« It seems to understand that the T cell response tests were carried out at the time of the diagnosis of Covid 19 to patients still on non-reduced therapy. It would be interesting to know if immunosuppression was reduced in all patients and with which drug (MMF?). In addition it would be interesting if this was associated with a different evolution from infection to disease »

We thank the reviewer for his interesting comment. The reviewer is right, T cell response tests were carried out at the time of the diagnosis of Covid 19 to patients still on non-reduced therapy. Baseline immunosuppression was modified after inclusion in 21 (47%) patients: antimetabolites were stopped in 20 patients and calcineurin inhibitors were stopped in one patient. This important information has been added to the revised version of the manuscript (results section, line 136-138). We did not found association between reduction of immunosuppression and outcomes probably because of the low number of patients.

« It would be interesting to isolate the population that did not take basicly antimetabolites (3 in the LOW group and 5 in the HIGH group) and look not only at the T cell response but if this best response correlates with the best outcome of the Covid infection »

We agree with the reviewer that this is a very interesting point. In multivariate analysis, the variable “no antimetabolite” significantly correlated with best outcome. All of these 8 patients had a good outcome (mild COVID, no pneumonia).

« It is not described if there have been deaths or loss of the graft (for example due to thrombophilic problems) »

There were 1 death, 2 ICU admissions and no graft loss. This information is mentioned in the results section of the revised manuscript: “Among the 45 patients of the study, 11 (24%) subsequently progressed to severe pneumonia requiring hospitalization and high dose dexamethasone, of whom two were transferred to an intensive care unit and one died. There was no graft loss.” (line 138-141).